# Functionally impaired isoforms regulate TMPRSS6 proteolytic activity

**Sébastien P. Dion**[1,2], **Antoine Désilets**[1,2], **Gabriel Lemieux**[1,2], **Richard Leduc**[1,2]*

**1** Department of Pharmacology-Physiology, Faculty of Medicine and Health Sciences, Université de Sherbrooke, Sherbrooke, Québec, Canada, **2** Institut de Pharmacologie de Sherbrooke, Faculty of Medicine and Health Sciences, Université de Sherbrooke, Sherbrooke, Québec, Canada

* Richard.Leduc@USherbrooke.ca

## Abstract

TMPRSS6 is a type II transmembrane serine protease involved in iron homeostasis expressed as 4 isoforms in humans. TMPRSS6 isoform 2 downregulates hepcidin production by cleaving hemojuvelin and other surface proteins of hepatocytes. The functions of catalytically impaired isoforms 3 and 4 are still unknown. Here we demonstrate that TMPRSS6 isoforms 3 and 4 reduce the proteolytic activity of isoform 2 and uncover the ability of isoforms to interact. Moreover, we identified 49 potential protein partners common to TMPRSS6 isoforms, including TfR1, known to be involved in iron regulation. By co-expressing TMPRSS6 and TfR1, we show that TfR1 is cleaved and shed from the cell surface. Further, we demonstrate that TMPRSS6 isoforms 3 and 4 behave as dominant negative.

## Introduction

The type II transmembrane serine protease TMPRSS6, also called matriptase-2, is mainly expressed in the liver [1] as a zymogen form that undergoes autoactivation at the cell membrane [2]. TMPRSS6 acts as a negative transcriptional regulator of *HAMP*, the gene coding for preprohepcidin, a protein precursor containing a 25 amino acid hormonal peptide, hepcidin, that regulates iron homeostasis [3]. Hepcidin functions by binding to the iron exporter protein ferroportin, which results in its internalization and degradation thereby decreasing iron delivery to plasma [4]. In hepatocytes, TMPRSS6 down regulates *HAMP*/hepcidin by interacting with and cleaving hemojuvelin and other essential components of the BMP-SMAD signaling pathway necessary to produce hepcidin [5, 6]. Hence, TMPRSS6 reduces BMP-SMAD signaling, preventing the overproduction of hepcidin and insufficient iron plasma levels. Supporting TMPRSS6's crucial role in iron regulation, it was discovered that mutations in the *TMPRSS6* gene can impair TMPRSS6 functionalities such as its autoactivation [7] which resulted in sustained BMP-SMAD signaling, elevated production of hepcidin, low plasma iron levels and thus iron-refractory iron deficiency anemia [8].

Because TMPRSS6 is a key modulator of hepcidin, down-regulating or inhibiting TMPRSS6 was first proposed as a therapeutic strategy to increase hepcidin levels in iron overload diseases such as hemochromatosis and β-thalassemia, two conditions for which TMPRSS6 has been validated as a pharmacological target [9–12]. Moreover, down regulation

**Data Availability Statement:** All relevant data are within the paper and its Supporting information files.

**Funding:** This work was supported by a grant from the Canadian Institutes of Health Research to R.L. (CIHR #PJT – 175271).

**Competing interests:** The authors have declared that no competing interests exist.

of TMPRSS6 function, which would increase hepcidin plasma levels, could be beneficial in other conditions with unmet medical needs such as infection with siderophilic pathogens [13], polycythemia vera [14] and obesity [15]. TMPRSS6 is therefore an attractive cell-surface target whose inhibition will increase circulating hepcidin levels and consequently impact on disease states related to hepcidin deficiency. From a pharmacological point of view, targeting cell-surface proteases to control their activity can be achieved through different strategies. Direct protease inhibitors of TMPRSS6 have been proposed by many groups, including ours [16, 17]. Another strategy that is currently used is to genetically knock down the expression of *TMPRSS6* transcripts using RNA-mediated interference (RNAi) [10, 18, 19] and antisense oligonucleotides (ASO) [9, 20].

Previously, we reported on the expression and characterization of 4 different human TMPRSS6 isoforms [1, 21]. Isoforms 1 (expressed at low levels) and 2 (the major isoform) are both enzymatically active. Transcripts encoding catalytically inactive entities of this protease were also described for the first time: isoform 3, containing an elongated exon 10 originating from an alternative polyadenylation signal, and isoform 4 which contains an insertion of 22 amino acids in the catalytic domain originating from alternative splicing (exon inclusion). We found that both protein-coding transcripts are expressed in hepatocellular carcinoma cell lines (HCC) [21], and isoform 4 is also expressed in normal liver [1]. While our studies predicted TMPRSS6 isoform expression based on transcriptional evidence, a deep proteome study has since identified isoform 4 specific peptides in human livers, supporting its expression at the protein level [22]. Until now, little is known about the biology of TMPRSS6 isoforms 3 and 4; the regulation at both the transcript and protein level, the identity of their protein partners as well as their physiological relevance related to TMPRSS6 biological functions all remains to be elucidated. Of note, we established that despite the fact that TMPRSS6 isoform 3 lacks a catalytic domain and that TMPRSS6 isoform 4 presents an insertion near its proteolytic serine, both isoforms can be expressed on the cell surface even though their proteolytic functions are altered [1]. Of note, we also established the differences between mouse and human TMPRSS6 isoforms and found no murine equivalent of TMPRSS6 isoforms 3 (truncated) and 4 [1]. Interestingly, *in vivo* studies previously reported a non-proteolytic role of mice TMPRSS6 ectodomains on hepcidin production [23], suggesting that TMPRSS6 isoforms 3 and 4 could potentially also be involved in iron regulation despite their altered proteolytic functions.

We have previously shown that TMPRSS6 isoforms 3 and 4 could interact with hemojuvelin and prevent its cleavage by TMPRSS6 active isoform 2 [1]. Herein, we show the dominant-negative property of TMPRSS6 isoforms 3 and 4 on TMPRSS6 function. In HEK293 cells, we show the negative impact of inactive isoforms expression on TMPRSS6 proteolytic activity. Importantly, we also demonstrate that TMPRSS6 isoforms can form homo- and hetero-interactions. Finally, we determine the interactome for TMPRSS6 isoforms 2, 3, and 4 and demonstrate that transferrin receptor 1 (TfR1) can be cleaved by TMPRSS6.

## Material and methods

### Cells, antibodies, and reagents

HEK293 and HepB cells were purchased from American Type Culture Collection (Manassas, VA). Hek293 cells were maintained in high glucose Dulbecco's Modified Eagle's Medium (DMEM) and Hep3B were cultured in Eagle's Minimum Essential Medium (EMEM), both medium were supplemented with 2 mM L-glutamine, 100 IU/ml penicillin, 100 μg/ml streptomycin and 10% foetal bovine serum. HCELL-100 was acquired from WISENT (St-Bruno, Canada). Anti-V5 antibody was purchased from Invitrogen (Waltham, MA). Anti-HA antibody was purchased from Roche (Mannheim, Germany). HRP-linked Anti-GAPDH rabbit

monoclonal antibody was purchased from Cell Signaling Technology (Danvers, MA). t-butox-ycarbonyl-Gln-Ala-Arg-7-amino-4-methylcoumarin (Boc-QAR-AMC) was purchased from R&D Systems (Minneapolis, MN). Lipofectamine 3000 was purchased from Invitrogen (Carlsbad, CA). Centrifugal filters were purchased from Merck Millipore (Cork, Ireland). Lysis buffer (1% Triton, 50 mM Tris, 150 mM NaCl, 5 mM EDTA) was supplemented with protease inhibitors from Roche (Mannheim, Germany). Protein A/G PLUS-agarose beads were purchased from Santa-Cruz Biotechnology (Dallas, TX). Anti-V5-tag mAb magnetic beads were purchased from MBL (Woburn, MA). Pierce MS-Grade trypsin was purchased from Thermo Fisher Scientific (Waltham, MA).

## Plasmid construction

Constructs encoding V5-tagged TMPRSS6-2, -3 and -4 were obtained and cloned as previously described [1]. TMPRSS6 HA-tagged isoforms were cloned from V5 constructs. TfR1-HA tagged construct was modified from a TfR1-V5 tagged construct obtained from Dr. Nabil G. Seidah's group.

## Proteolytic activity of TMPRSS6 isoforms

HEK293 cells were transfected with 2 μg of DNA using Lipofectamine 3000 in 6-well plates. 1 μg was transfected for each individual TMPRSS6 isoforms. Twenty-four hours post-transfection, cell media was changed for HCELL-100 media for an additional 24 hours. Proteolytic activity was measured by monitoring Boc-QAR-AMC cleavage using a FLx800 TBE microplate reader (Bio-Tek Instruments, Winooski, VT) as previously described [1, 24, 25]. Cell lysates were analyzed on SDS-Polyacrylamide gels (10%) and blotted with an anti-V5 and anti-GAPDH antibodies.

## TMPRSS6 isoforms homo- and hetero-interactions

HEK293 cells were co-transfected with a total of 2 μg of DNA. When a TMPRSS6 isoform was expressed alone, only 1 μg of encoding DNA was transfected alongside 1 μg of empty vector DNA. At 24 hours post-transfection, cells were washed, and lysis realized at 4˚C. Protein samples (250 μg) were immunoprecipitated in 500 μL volume for 24 hours at 4˚C with an anti-V5 antibody and Protein A/G PLUS-agarose beads. Immunoprecipitated proteins and cell lysates were loaded on SDS-polyacrylamide gels and analyzed by immunoblotting using anti-HA, anti-V5 and GAPDH antibodies.

## Immunoprecipitation of TMPRSS6 isoforms interacting partners

Hep3B cells were transfected in 10 cm plates with 10 μg of pcDNA6-V5 empty vector or TMPRSS6 isoforms 2, 3 or 4 V5-tagged constructs. 48 hours post-transfection, membrane preparations of cells were executed as previously described [26]. Immunoprecipitations were carried on membrane preparations using anti-V5-tag mAb magnetic beads. Buffers were prepared in MS-grade water and low-binding microtubes were used for all mass spectrometry sample preparation steps.

## Protein digestion, purification and desalting of peptides

Following washing of the beads with 20 mM $NH_4HCO_3$, immunoprecipitated proteins were digested on beads using Pierce MS-Grade trypsin for at least 5h at 37˚C. Digestion was stopped by adding 1% formic acid prior to centrifuging at 2000g for 3 minutes. The supernatant was conserved and beads were resuspended in 100 μL of a 60% $CH_3CN$, 1% formic acid solution

for 5 minutes with shaking. A second centrifugation was performed, and supernatant pooled with the first one obtained. Samples were then dried in a speed vac, resuspended in 30 μL of 0.1% trifluoroacetic acid and desalted using a ZipTip as previously described [27].

### LC-MS/MS analysis

Mass spectrometry analysis of peptides was carried by the Université de Sherbrooke Proteomic Platform. Digested peptides separation was done using a Dionex Ultimate 3000 nanoHPLC system coupled to an OrbiTrap QExactive mass spectrometer via an EasySpray Source as previously described [27].

### Protein identification and fold-change enrichment analysis

Raw files were analyzed with MaxQuant software (version 1.5.2.8) and Uniprot human proteome database. For MaxQuant analysis, trypsin was selected as the enzyme used (K/R, not before P), label-free-quantification was used, match between runs were not allowed and the minimum peptide count was set to 1. Identification values (PSM FDR, Protein FDR and Site decoy fraction) were set to 0.05. Fold change enrichments versus mock control samples were calculated using MS/MS count in CRAPome [28].

### TfR1 cleavage by TMPRSS6 isoforms

Hep3B cells were co-transfected with 1 μg of TMPRSS6-2 and 1 μg of TfR1 constructs using Lipofectamine 3000 in 6 well plates. 24 hours post-transfection, cell media was changed for HCELL-100. The next day, 1 mL of cell media was collected and concentrated before cell lysis was performed. Equal amount of cell lysate (30 μg) and volume of concentrated media (30 μL) were loaded on SDS-polyacrylamide gels and analyzed with immunoblotting using anti-HA, anti-V5 and anti-GAPDH antibodies. Band intensities were quantified by densitometry analysis using Bio-1D software (Vilber Lourmat) and compared to the cleavage by TMPRSS6-2 alone (100%).

## Results

### TMPRSS6 isoforms 3 and 4 reduce isoform 2 activity

In humans, the *TMPRSS6* gene leads to the transcription of 4 known, annotated transcripts predicted to be expressed as proteins (Fig 1A) [1, 29]. Since TMPRSS6 isoform 1 (TMPRSS6-1) is expressed at low levels [1, 21], we studied the predominantly expressed and proteolytically active TMPRSS6 isoform 2 (TMPRSS6-2). To evaluate whether TMPRSS6 isoforms 3 and 4 (TMPRSS6-3, TMPRSS6-4) have a direct impact on TMPRSS6-2 activity, we co-expressed either one of TMPRSS6 inactive isoforms with active TMPRSS6-2 in HEK293 cells. First, we verified C-terminal V5-tagged TMPRSS6 isoforms expression in transfected cell lysates by Western blotting using an antibody directed against V5 (Fig 1B). As expected, TMPRSS6-2 and TMPRSS6-4 are detected as ~100 kDa bands while TMPRSS6-3 is detected as a ~60 kDa band (Fig 1B, upper panel). The proteolytic activity of TMPRSS6 isoforms in the extracellular medium of HEK293 transfected cells was measured using a fluorogenic substrate (Fig 1C). As previously described, enzymatic activity was detected for TMPRSS6-2, but not TMPRSS6-3 and TMPRSS6-4 [1]. When the same quantity of plasmid coding for active TMPRSS6-2 was co-transfected alongside either one of TMPRSS6 inactive isoforms 3 or 4, a significant reduction in relative activity was detected, supporting our hypothesis that TMPRSS6-3 and TMPRSS6-4 behave as dominant negative isoforms.

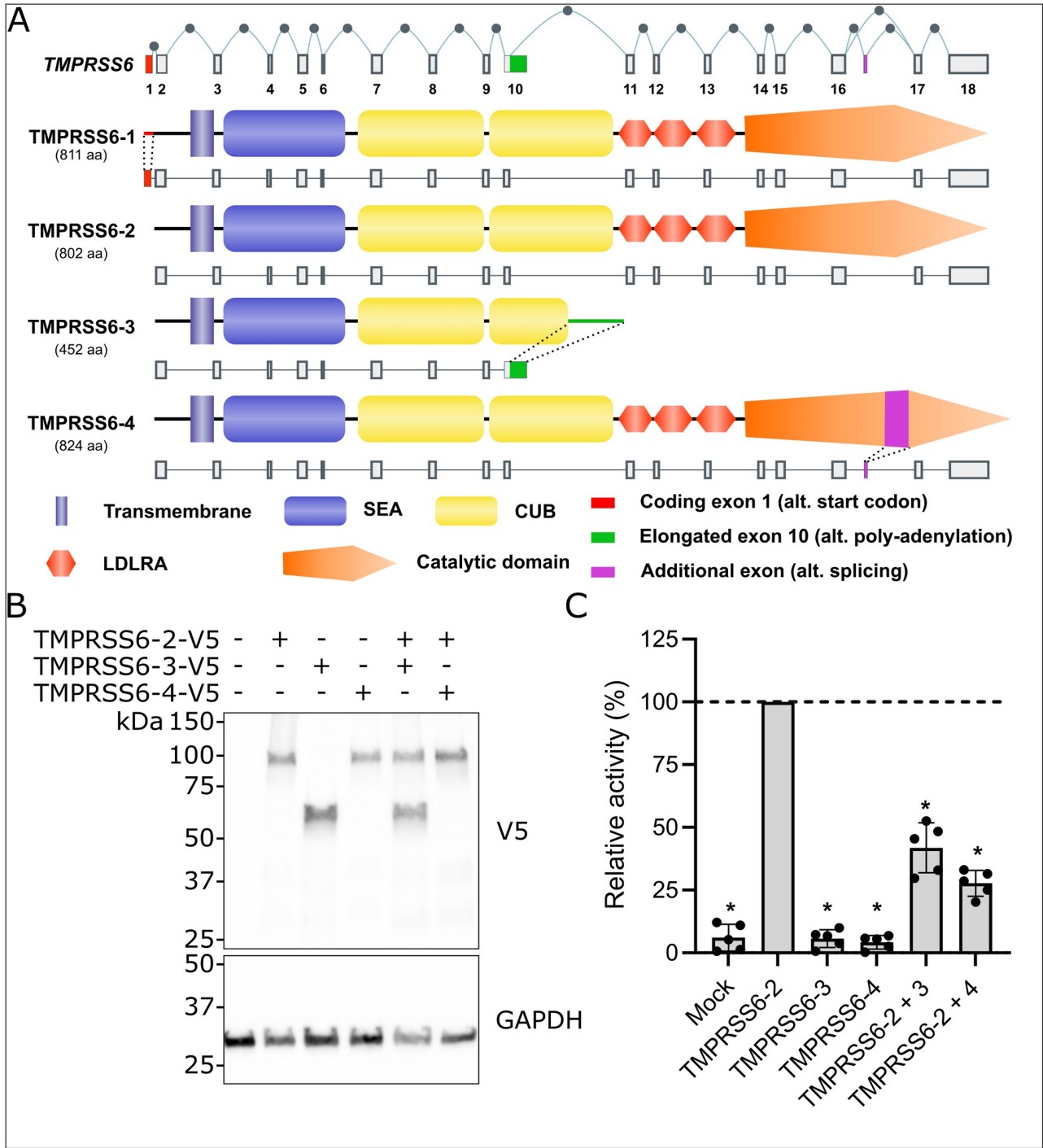

**Fig 1. TMPRSS6 isoforms 3 and 4 reduce isoform 2 activity.** A) TMPRSS6 isoforms representation adapted from our previous publication [1] licensed under CC BY 3.0. B) Expression of TMPRSS6 V5-tagged isoforms 2, 3 and 4 in transfected HEK293 cells assessed by western blotting against V5. C) Proteolytic activity was measured in the cell medium of HEK293 cells transfected with one or two TMPRSS6 V5-tagged isoforms. Boc-QAR-AMC (200 μM) fluorogenic substrate cleavage was monitored. Results are presented as relative activity over TMPRSS6 isoform 2 (TMPRSS6-2) activity. Statistical significance was assessed using one sample T test, $^*p < 0.0002$ (n = 5).

## TMPRSS6 isoforms engage in homo- and hetero-interactions

Because inactive isoforms had a direct impact on TMPRSS6-2 activity, we next verified if TMPRSS6 isoforms could interact together. To investigate this possibility, we proceeded to perform co-immunoprecipitation experiments. We co-transfected C-terminal V5- and HA-tagged TMPRSS6 isoforms in HEK293 cells, alone or in combination, and proceeded to the immunoprecipitation of V5-tagged proteins (Fig 2). By blotting using an antibody directed against the V5 epitope, we confirmed immunoprecipitations for all conditions in which TMPRSS6 V5-tagged isoforms were expressed (Fig 2, second panel, lanes 2–4 and 8–12). Interestingly, when TMPRSS6 V5- and HA-tagged isoforms were co-expressed, we could detect HA signal, indicating an interaction between TMPRSS6 isoforms. These interactions were also detected when the same isoforms with different tags were co-expressed (Fig 2, upper panel, lanes 8–10) but also when TMPRSS6-2 was co-expressed with TMPRSS6-3 or TMPRSS6-4 (Fig 2, upper panel, lanes 11 and 12).

Hence, using this technique, we demonstrate for the first time that TMPRSS6 isoforms 2, 3 and 4 can individually interact with an identical isoform (homo-interaction) and that isoform

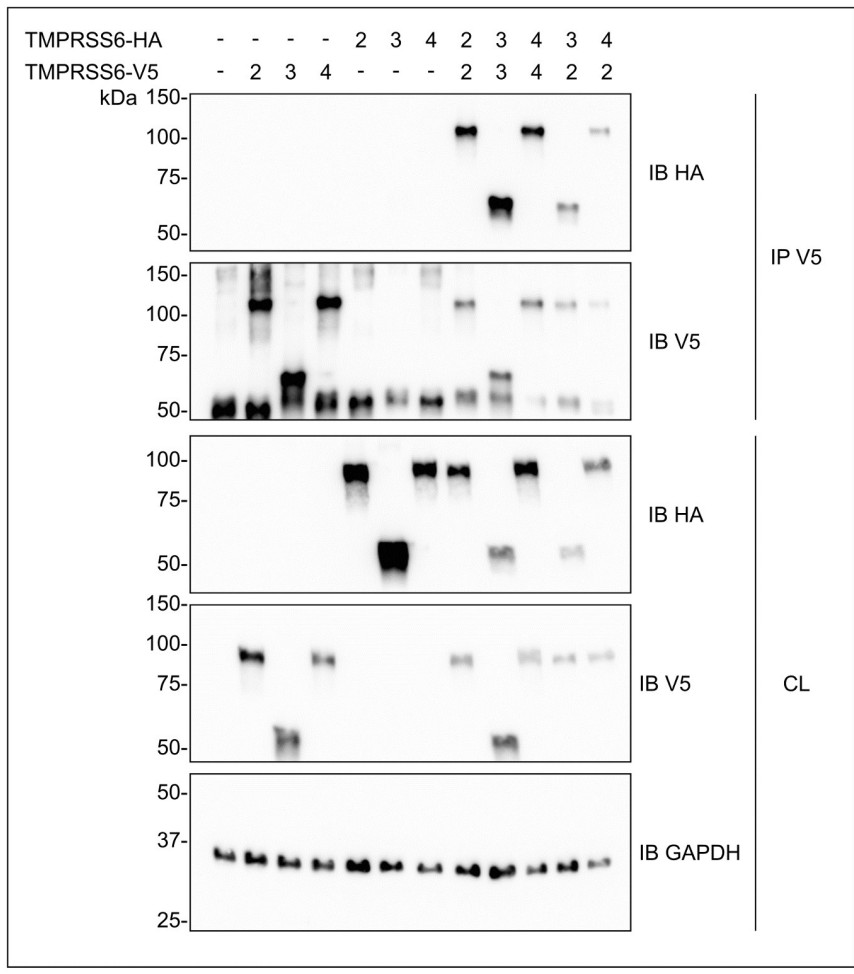

**Fig 2. TMPRSS6 isoforms engage in homo- and hetero-interactions.** HEK293 cells were co-transfected with V5- and HA-tagged TMPRSS6 isoforms. Homo- and hetero-interaction of TMPRSS6 isoforms were assessed by immunoprecipitation (IP) of cell lysate (CL) using an anti-V5 antibody. Isoform immunoprecipitation was detected using anti-V5 and anti-HA antibodies. Equal amount of CL was loaded on SDS-polyacrylamide gels and cell GAPDH was used as a loading control (n = 3).

2 could also interact with isoforms 3 and 4 (hetero-interactions). These results suggest that TMPRSS6 exist as dimers or in protein complexes and importantly that interaction of inactive isoforms 3 and 4 with the major TMPRSS6 isoform, TMPRSS6-2, could be a mechanism to regulate its function.

## Identification of TMPRSS6 isoforms interacting partners

To gain further insight in the biological functions of TMPRSS6, we next generated an interactome signature profile of TMPRSS6. For this purpose, we selected the commonly used hepatocellular carcinoma cell line Hep3B, known to express TMPRSS6 and several components of the iron regulation pathway [21] as a model system. Following transfection of these cells with TMPRSS6 V5-tagged isoforms 2, 3 and 4, we proceeded to membrane isolation and TMPRSS6 immunoprecipitation. Interacting proteins were then identified by mass spectrometry and their abundance analyzed. Proteins with an enrichment fold change $\geq 3$ versus mock-transfected cells were considered as potential binding partners of TMPRSS6 isoforms. Using this technique, we were identified 99 potential protein partners for TMPRSS6-2, 154 for TMPRSS6-3 and 164 for TMPRSS6-4 (S1 Table for complete list).

By protein clustering between the different isoforms using a Venn diagram, a total of 233 unique potential protein partners were assigned in 7 different groups (Fig 3A) [30]. To limit our proteomic analysis for the most relevant and robust protein partners, we focused on the 49 binding partners that were common to all isoforms. The functional interactome for these candidates was then revealed using STRING (Fig 3B) [31]. Amongst these, we found the bait-TMPRSS6 (Fig 3B, top left) as well as 40 other membrane-associated proteins (blue nodes) and 6 proteins involved in the regulation of lipid biosynthetic process (red nodes). Three proteins which already had a curated association with TMPRSS6 were identified: TUBGCP3 (tubulin gamma complex associated protein 3), APOB (apolipoprotein B-100) and TfR1 (transferrin receptor 1). While APOB is reported to be co-expressed with TMPRSS6, both TUBGCP3 and TfR1 are associated with TMPRSS6 due to text mining evidence and importantly, the latter is involved in iron homeostasis. Overall, these results led us to identify new potential partners for TMPRSS6 and open the possibility to refine our understanding of all TMPRSS6 isoforms.

## TMPRSS6 cleaves TfR1

Because both TMPRSS6 and TfR1 are involved in iron homeostasis mechanisms, the latter being responsible for cellular iron uptake [32], we found this novel interaction particularly relevant to study. We speculated whether TfR1 could be cleaved by TMPRSS6 since it had previously been shown that transferrin receptor 2 (TFR2), sharing 45% identity with TfR1 in humans, was indeed cleaved by TMPRSS6 using mouse orthologues [5].

To assess the ability of TMPRSS6 to cleave TfR1, we co-transfected V5-tagged TMPRSS6 isoforms with HA-tagged TfR1 in Hep3B cells and analyzed both cell lysates and conditioned media by immunoblotting (Fig 4). Expression of TfR1 and TMPRSS6 isoforms was confirmed in the cell lysates (Fig 4, three upper panels). When TfR1 is co-expressed with active TMPRSS6-2, shedding of TfR1 was detected in the concentrated cell media as a ~70 kDa band (Fig 4A, lower panel, lane 3). This proteolytic event was not detected when TfR1 and TMPRSS6-3 and TMPRSS6-4 were co-expressed. However, TMPRSS6 isoforms with impaired activities reduced TfR1 cleavage by TMPRSS6-2 as shown by densitometry quantification (Fig 4A, lower panel, lanes 5 and 7 and Fig 4B). Mutation of TMPRSS6 isoform 2 catalytic serine to alanine (S762A) abrogated TfR1 cleavage while co-transfection of TfR1 with the proprotein convertase 7 (PC7) did lead to TfR1 shedding (S1 Fig). PC7 was previously shown to be able to cleave TfR1 at the $Arg_{100}$ site [33]. We observed a cleavage product of the same

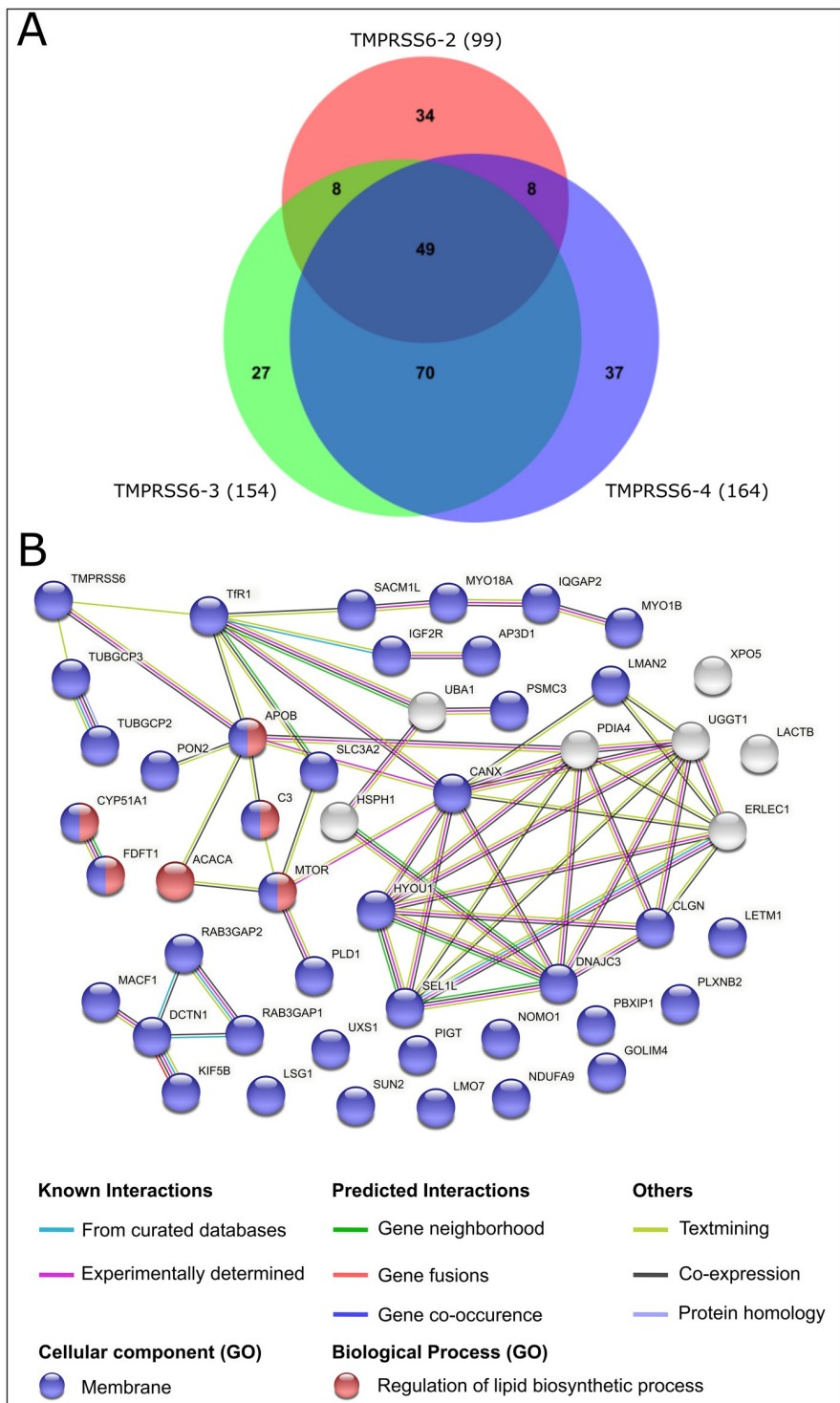

**Fig 3. Identification of TMPRSS6 isoforms interacting partners.** A) BioVenn overlap of TMPRSS6 isoforms 2, 3 and 4 interacting partners (fold change vs mock ≥ 3) identified by mass spectrometry analysis of TMPRSS6 V5-tagged immunoprecipitation from transfected Hep3B cell membrane preparations. B) STRING interactome of 49 common protein partners.

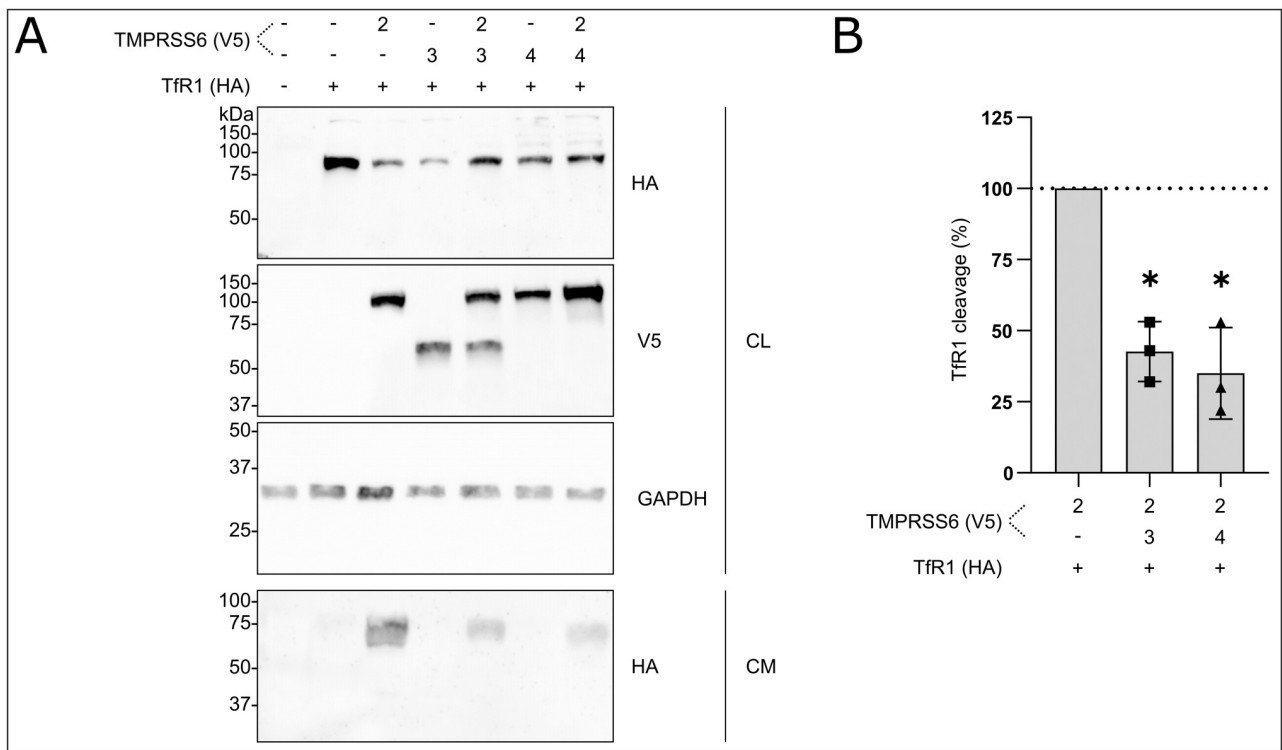

**Fig 4. TMPRSS6 cleaves TfR1.** A) TfR1-HA was transfected with one or two TMPRSS6-V5 tagged isoforms. Expression was detected in the cell lysate (CL) and cell media (CM) (n = 3). B) TfR1 cleavage quantification. Results are presented as TFR1 cleavage (%) relative to TFR1 cleavage by TMPRSS6 isoform 2. Statistical significance was assessed using one sample T test, *p < 0.02. The means ± SD are presented (n = 3).

molecular weight for both TMPRSS6 and PC7, suggesting that TMPRSS6 could potentially also cleave at the $Arg_{100}$ site or to a neighbouring residue (S1 Fig).

These results indicate that TfR1 could be a TMPRSS6 substrate and suggest that its cleavage could be modulated via functionally impaired isoforms TMPRSS6-3 and TMPRSS6-4. However, validation of TfR1 cleavage by TMPRSS6 remains to be confirmed using more physiologically relevant human cellular models such as primary hepatocytes, induced pluripotent stem cells derived from hepatocytes or organoids.

## Discussion

The type II transmembrane serine protease TMPRSS6 is an important factor involved in iron homeostasis. Under certain pathological conditions such as hemochromatosis, beta-thalassemia, and polycythaemia vera, an increase in hepcidin would lower the iron burden of patients and thus would be beneficial to patients, making TMPRSS6 an appealing therapeutic target to treat iron overload disorders [34]. Moreover, a reduction of TMPRSS6 levels has been described as protective against obesity in direct relation with its role in iron homeostasis [15]. Despite the great interest in the scientific community for TMPRSS6 pathophysiological functions, little is known with regards to the role and relevance of the distinct isoforms.

We show in this study that TMPRSS6 inactive isoforms 3 and 4 behave as dominant negative proteins affecting TMPRSS6 major isoform's catalytic activity and substrate cleavage efficiency. We demonstrate by co-immunoprecipitation experiments that TMPRSS6 isoforms can form homo- and hetero-interactions, providing insights into the mechanism leading to

reduced activity. Formation of homo-interaction is supported by a previous study in which TMPRSS6 catalytic domain is found in dimeric form in conditioned media [6]. Interestingly, homodimerization has been well characterized for matriptase [35], a type II transmembrane serine protease expressed in most epithelial cells [36], that shares the same domain composition as TMPRSS6, except that it harbours 3 instead of 4 LDLRA domains [37]. Indeed, a detailed biochemical characterization demonstrates that matriptase could homodimerize as an inactive transient complex when in the absence of its cognate inhibitor hepatocyte growth activator inhibitor-1 (HAI-1) [35]. More studies will be needed to investigate the formation of TMPRSS6 homo- and hetero-interactions and the biological relevance of these interactions. However, one could envision the possibility that homodimerization is the normal process for TMPRSS6 autoactivation and heterodimerization with inactive isoforms, in addition to substrate sequestration, a refined mechanism by which these inactive moieties regulate TMPRSS6 function.

Our experiments also provide new information regarding TMPRSS6 interacting proteins. Using mass spectrometry, we identified 232 novel potential binding proteins of TMPRSS6. Of these, 49 proteins (including TMPRSS6 itself) were common to TMPRSS6 isoforms 2, 3 and 4. Hemojuvelin (HJV), a known TMPRSS6 substrate [6], was not identified in our pull-down assays possibly due to the low expression of *HFE2*, HJV encoding gene, which we have previously shown to be significantly less expressed in Hep3B cells than in the human liver [21].

The interactome analysis allowed the identification of proteins involved in lipid metabolism such as the apolipoprotein B-100 (APOB) and acetyl-CoA carboxylase 1 (ACACA). Interestingly, the latter was described as overexpressed in Tmprss6$^{-/-}$ mice, which are resistant to diet-induced obesity [15].

Amongst these 49 common proteins, we also found transferrin receptor 1 (TfR1). This interaction is of particular interest since a genome-wide association study previously reported a correlation between soluble TfR1 levels and the presence of single nucleotide polymorphisms in *PCSK7* (rs236918) and *TMPRSS6* (rs855791) genes [38]. In the hepatocellular carcinoma cell line Huh7, it was previously demonstrated that TfR1 could be proteolytically cleaved by PC7, encoded by *PCSK7*, but not by TMPRSS6 [33]. However, by co-expressing TfR1 and TMPRSS6 isoforms in Hep3B cells, we show here that active TMPRSS6 isoform 2 cleaves this receptor leading to cell surface shedding. We also demonstrate that TMPRSS6 isoforms 3 and 4 reduce TfR1 cleavage by TMPRSS6 isoform 2 in a dominant negative manner, similarly to what was observed for hemojuvelin [1]. Of note, *TMPRSS6* SNP rs855791 encodes for valine to alanine change at position 736 within the proteolytic domain of TMPRSS6 isoform 2. Interestingly, this variant, which is found in Hep3B cells[18] is associated with higher susceptibility to hepatic iron accumulation in thalassemia patients [39] and lower hepcidin levels in healthy individuals [40]. These results suggest a gain-of-function for the V736A variant, that could lead to increased TfR1 cell surface shedding, but the molecular mechanism involved still needs to be determined.

Taken together, the results presented in this study highlight the importance of studying the distinct TMPRSS6 isoforms. We demonstrated that TMPRSS6 isoforms could engage in homo- and hetero-interactions altering the enzyme's proteolytic activity and substrate cleavage efficiency. We also identified potential novel interactors common to TMPRSS6 isoforms and confirmed that one of these, transferrin receptor 1 (TfR1), is a potential TMPRSS6 substrate that still needs to be validated. Even though no clear function has been associated with TfR1 cleavage so far, the measurement of soluble TfR1 in serum is used to discriminate between different types of anaemias [41]. Understanding the physiological impact of TfR1 cleavage by TMPRSS6 could lead to a better comprehension of this protease's fine-tuning function in iron homeostasis.

Next steps would include confirming interaction of TMPRSS6 isoforms with other potential partners and to elucidate the consequence of such interactions. Importantly, our work supports the need to specifically unravel the biological functions of isoform 3 and 4 in a physiological context.

## Supporting information

**S1 Fig. TfR1 cleavage controls.** TfR1-HA was transfected either with V5-tagged TMPRSS6 isoform 2 WT (active, TMPRSS6-2-WT-V5), catalytically inactivated TMPRSS6 isoform 2 (TMPRSS6-2-S762A-V5) or proprotein convertase 7 (PC7-V5). Expression was detected in the cell lysate (CL) and cell media (CM) (n = 3).
(TIF)

**S1 Table. Interacting protein list.**
(TIF)

## Acknowledgments

S.P.D. is a Frederick Banting and Charles Best Canada Graduate Scholarships Doctoral Awards fellow. We would like to thank the Université de Sherbrooke Proteomic Platform for their expertise and Dr. Nabil Seidah for providing the TfR1 DNA construct.

 S.P.D. and G.L. performed the experiments.

 S.P.D., A.D., and R.L. conceptualized the study.

 S.P.D., A.D., G.L. and R.L. wrote the original manuscript.

 R.L. acquired funding.

## Author Contributions

**Conceptualization:** Sébastien P. Dion, Antoine Désilets, Richard Leduc.

**Data curation:** Sébastien P. Dion, Gabriel Lemieux.

**Funding acquisition:** Richard Leduc.

**Investigation:** Sébastien P. Dion, Antoine Désilets.

**Methodology:** Sébastien P. Dion, Antoine Désilets.

**Project administration:** Antoine Désilets.

**Visualization:** Sébastien P. Dion.

**Writing – original draft:** Sébastien P. Dion.

**Writing – review & editing:** Sébastien P. Dion, Antoine Désilets, Gabriel Lemieux, Richard Leduc.

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
