## [Decision Letter · Decision Letter 0]

17 Aug 2022

Functionally impaired isoforms regulate TMPRSS6 proteolytic activity

PONE-D-22-19267

Dear Dr. Leduc,

We’re pleased to inform you that your manuscript has been judged scientifically suitable for publication and will be formally accepted for publication once it meets all outstanding technical requirements.

Kind regards,

Matthew Bogyo, Ph.D.

Academic Editor

PLOS ONE

1. PLOS requires an ORCID iD for the corresponding author in Editorial Manager on papers submitted after December 6th, 2016. Please ensure that you have an ORCID iD and that it is validated in Editorial Manager. To do this, go to ‘Update my Information’ (in the upper left-hand corner of the main menu), and click on the Fetch/Validate link next to the ORCID field. This will take you to the ORCID site and allow you to create a new iD or authenticate a pre-existing iD in Editorial Manager. Please see the following video for instructions on linking an ORCID iD to your Editorial Manager account: https://www.youtube.com/watch?v=_xcclfuvtxQ

Reviewers' comments:

Reviewer's Responses to Questions

**Comments to the Author**

1. Is the manuscript technically sound, and do the data support the conclusions?

Reviewer #1: Yes

2. Has the statistical analysis been performed appropriately and rigorously? 

Reviewer #1: I Don't Know

3. Have the authors made all data underlying the findings in their manuscript fully available?

Reviewer #1: Yes

4. Is the manuscript presented in an intelligible fashion and written in standard English?

Reviewer #1: Yes

5. Review Comments to the Author

Reviewer #1: The paper outlines a nice discovery of the function of several TMPRSS6 isoforms 2, 3, and 4. Also multiple binding partners to these isoforms have been identified, one of them being TfR1, a known regulator of iron in common with TMPRSS6/hepcidin. The authors have previously described the discovery and expression of all these isoforms but the new data and results presented here provide insight into their biological role(s) and are the oucome compelling with th edata and analysis of high scientific merit and quality. While these studies provide a limited contribution to the field, the ultimate relevance of these isoforms 2-4 in pathphysiological conditions such as iron overload or deficiency is not demonstrated in vivo (animals or humans). However, this preliminary work is acceptable for communication in PLOS One.

6. PLOS authors have the option to publish the peer review history of their article (what does this mean?). If published, this will include your full peer review and any attached files.

Reviewer #1: No

---

## [Editor Report · Acceptance letter]

22 Aug 2022

PONE-D-22-19267 

Functionally impaired isoforms regulate TMPRSS6 proteolytic activity 

Dear Dr. Leduc:

I'm pleased to inform you that your manuscript has been deemed suitable for publication in PLOS ONE. Congratulations! Your manuscript is now with our production department. 

Kind regards, 

on behalf of

Dr. Matthew Bogyo 

Academic Editor

PLOS ONE